# LLAVIDAL : Benchmarking Large LAnguage VIsion Models for Daily Activities of Living

**Rajatsubhra Chakraborty**\*  **Arkaprava Sinha**\*  **Dominick Reilly**\*  **Manish Kumar Govind**
**Pu Wang**  **Francois Bremond**†  **Srijan Das**

UNC Charlotte   †Inria   †Université Côte d'Azur

\* Equal contribution   {rchakra6, asinha13, dreilly1}@charlotte.edu

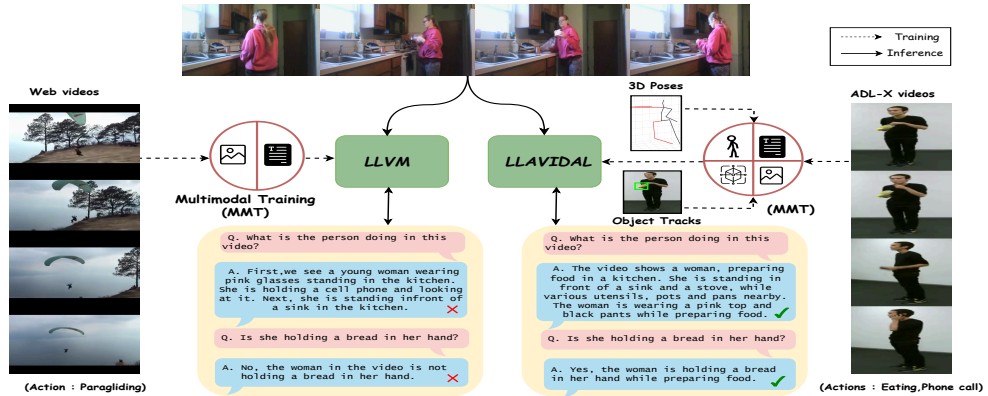

Figure 1: **Comparison of LLVM vs LLAVIDAL** : In real world scenarios, web-video trained models struggle to understand Activities of Daily Living due to the subtle nuances in the video, whereas our **ADL-X** trained LLAVIDAL model triumphs in understanding complex human-object interactions.

## Abstract

Large Language Vision Models (LLVMs) have demonstrated effectiveness in processing internet videos, yet they struggle with the visually perplexing dynamics present in Activities of Daily Living (ADL) due to limited pertinent datasets and models tailored to relevant cues. To this end, we propose a framework for curating ADL multiview datasets to fine-tune LLVMs, resulting in the creation of **ADL-X**, comprising 100K RGB video-instruction pairs, language descriptions, 3D skeletons, and action-conditioned object trajectories. We introduce **LLAVIDAL**, an LLVM capable of incorporating 3D poses and relevant object trajectories to understand the intricate spatiotemporal relationships within ADLs. Furthermore, we present a novel benchmark, **ADLMCQ**, for quantifying LLVM effectiveness in ADL scenarios. When trained on ADL-X, LLAVIDAL consistently achieves state-of-the-art performance across all ADL evaluation metrics. Qualitative analysis reveals LLAVIDAL's temporal reasoning capabilities in understanding ADL. The link to the dataset is provided at: https://adl-x.github.io/

## 1   Introduction

Human cognitive perception integrates information from multiple sensory modalities to form a unified representation of the world [1]. Towards emulating human cognitive perception in digital intelligence, initial efforts focused on integrating vision and language modalities [2, 3, 4, 5, 6]. Subsequently,

Submitted to the 38th Conference on Neural Information Processing Systems (NeurIPS 2024) Track on Datasets and Benchmarks. Do not distribute.

the success of LLMs like GPT [7], PALM [8], BLOOM [9] led to the introduction of multimodal conversational models[10, 11, 12, 13, 14, 15, 16] that combine image pixels and LLMs, we dub as Large Language-Vision Language Models (LLVMs). However, these image-based LLVMs lack the capability for complex reasoning and interactions, particularly in understanding spatio-temporal relationships involved in human activities. In this study, we investigate the understanding of Activities of Daily Living (ADL) videos by LLVMs, which present various challenges including multiple exo-centric viewpoints, fine-grained activities with subtle motion, complex human-object interactions, and long-term temporal relationships. We envision that LLVMs capable of addressing these challenges will significantly influence the future intelligent systems, particularly in healthcare applications such as eldercare monitoring, cognitive decline assessment, and robotic assistance development.

Recently, [17, 18, 19, 20, 21, 22, 23] have integrated videos into LLMs, leading to the development of video-based LLVMs capable of capturing spatio-temporal features. However, these models are predominantly trained on large-scale web videos [24, 25, 26, 27, 28], which mainly consists of sports clips, movie excerpts, and instructional videos. These videos, typically filmed by professionals, follow strict temporal sequences in closely controlled background (e.g., Paragliding). The evident temporal structure and scene semantics in such videos facilitate spatial understanding within LLVMs, as shown in 1. In contrast, ADL videos pose additional challenges, characterized by temporal unstructuredness where diverse actions may unfold concurrently within a single sequence [29]. For instance, *a person cooking could intermittently engage in unrelated activities like making a phone call or drinking water, disrupting the linear progression of the composite action cooking*. Consequently, existing LLVMs trained on web videos struggle to capture such visually perplexing dynamics inherent in ADL scenarios. Moreover, unlike specialized video architectures designed for understanding ADL [30, 31, 32, 33, 34, 35, 36], these LLVMs lack explicit utilization of cues like 3D poses or object encodings, which are crucial for understanding ADL. These cues aid in learning view-invariant representations and capturing fine-grained details essential for interpreting complex human activities. Hence, the current limitations in understanding ADL stem from the lack of instruction tuning of LLVMs on real-world multiview ADL datasets captured in indoor settings and the simplistic design of LLVMs with holistic operations.

To this end, we propose a framework of curating ADL videos for instruction tuning LLVMs. This framework introduces the **ADL-X** dataset, comprising 100K untrimmed RGB video-instruction pairs, 3D poses (P), language descriptions, and action-conditioned object trajectories (see Table 1). We then introduce the **L**arge **LA**nguage **VI**sion model for **D**aily **A**ctivities of **L**iving (**LLAVIDAL**), trained on ADL-X, which integrates videos, 3D poses, and object cues into the LLM embedding space. Our study explores various strategies for integrating 3D pose information and human-object interactions within LLVMs, demonstrating that language contextualized features extracted from 3D poses and object trajectories can effectively be integrated into LLAVIDAL. Furthermore, we introduce a benchmark ADL Multiple Choices Question (**ADLMCQ**), specifically designed to evaluate the effectiveness of LLVMs for ADL. ADLMCQ includes action recognition (ADLMCQ-AR) and action forecasting (ADLMCQ-AF), assessed through a multiple choice question-answering task. We also evaluate existing LLVMs for generating video description of ADL scenes and compare their performance with LLAVIDAL. Our empirical findings indicate that LLAVIDAL with object cues, outperforms other LLVMs, including those trained on datasets of ten times the size, on the ADL benchmarks.

To summarize our contributions:

- We introduce ADL-X, the first multiview RGBD instruction ADL dataset, curated through a novel semi-automated framework for training LLVMs.
- LLAVIDAL is introduced as the first LLVM tailored for ADL, incorporating 3D poses and object cues into the embedding space of the LLM.
- A new benchmark, ADLMCQ, is proposed for an objective evaluation of LLVMs on ADL tasks, featuring MCQ tasks for action recognition & forecasting.
- Exhaustive experiments are conducted to determine the optimal strategy for integrating poses or objects into LLAVIDAL. Evaluation of existing LLVMs on ADLMCQ and video description tasks reveals that LLAVIDAL trained on ADL-X significantly outperforms baseline LLVMs.

Table 1: Video Instruction Dataset Comparison.

| Dataset | Modalities | Subjects | Multiple Views | Videos | QA Pairs | Atomic Actions per Vid | Temporal Rand. | Object Traj. | Type |
|---|---|---|---|---|---|---|---|---|---|
| TimeIT[21] | RGB+L | NA | No | 173000 | 173K | Medium | No | No | Web |
| VideoChat[17] | RGB+L | NA | No | 8196 | 11K | Low | No | No | Web |
| Valley[26] | RGB+L | NA | No | 64,687 | 65K | Low | No | No | Web |
| VideoChatGPT [20] | RGB+L | NA | No | 27,801 | 100K | Medium | No | No | Web |
| **ADL-X** | **RGB+P+L** | **106** | **Yes** | 16,343 | 100K | **High** | **Yes** | **Yes** | **ADL** |

## 2 Semi-automated Framework for generating ADL Video-instructions Pairs

This section describes the data curation framework employed for the creation of a novel dataset, ADL-X. This dataset specifically caters to the instruction tuning of LLVMs within the ADL domain. ADL-X comprises video recordings of ADLs. To enrich the dataset and facilitate LLM training, question-answer (QA) pairs were generated from a corpus of long-form ADL videos. These QA pairs target various aspects of the ADLs, including: human pose configuration, objects relevant to the human actions, scene appearance, and the fine-grained actions performed. We hypothesize that incorporating such instructional tuning during the LLVM training process will promote alignment of visual tokens within the LLM's embedding space. ADL-X represents a comprehensive ADL dataset encompassing various modalities: - RGB videos, 3D poses, Language descriptions, object tracklets. This rich dataset offers a valuable tool for evaluating the capabilities of LLVMs in tasks related to ADLs, including description, recognition, and anticipation.

A critical characteristic of ADL videos lies in the inherent spontaneity of the actions performed. Unlike scripted scenarios [25, 37, 38], fine-grained actions within ADLs often occur randomly. To capture this essential characteristic within our dataset, we curated ADL-X from NTU RGB+D 120 dataset [39]. This selection was motivated by the dataset's focus on ADL videos and its inherent diversity in terms of actions, subjects, and camera viewpoints. Also, this data curation framework could be extended to any existing trimmed/untrimmed ADL datasets [40, 41, 42]. Below, we elaborate the steps involved in building the ADL-X in a chronological order.

**Person-centric Cropping.** ADL tasks necessitate a focus on the individual performing the actions, the actions themselves, and the human-object interactions. To achieve this targeted focus within the data curation framework, we implemented a person-centric cropping strategy leveraging the pose information captured through Kinect sensors [43]. By using the pose information in each frame of the NTU RGB+D 120 dataset, we are able to detect and crop out the person(s) performing the actions. This cropping process effectively reduces the amount of background information present in the videos, eliminating data irrelevant to the target ADLs. This step is crucial as existing ADL datasets often contain extensive background information that is not relevant to the actions being performed. The presence of such extraneous information can significantly hinder subsequent stages within the data curation framework.

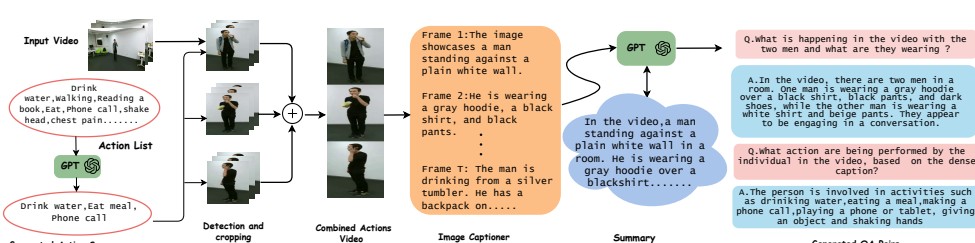

Figure 2: Dataset Curation Pipeline: We employ CogVLM[44] as our person-centric image captioner and GPT 3.5 Turbo[7] as our summarizer and QA generator.

**Stitching shorts clips.** To capture the inherent randomness of real-world ADLs, we constructed a set of 160 composite action sequences. These sequences were generated by prompting a GPT to combine individual actions from the original NTU RGB+D 120 dataset's list of 120 actions (denoted as $A_1$, $A_2$, ..., $A_{120}$). An example sequence structure could be represented as $A_1 \rightarrow A_3 \rightarrow A_{17}$. Following these

generated composite action sequences, we temporally stitched together short video clips ($clip_j^a$, where $a$ is the action class) from the NTU dataset. This stitching process ensured that all clips within a video belonged to the same subject and camera view, maintaining coherence in the resulting video sequence. For instance, a stitched video sequence might be represented as $[clip_{r1}^1 \quad clip_{r2}^3 \quad clip_{r3}^{17}]$ where $r1$, $r2$, $r3$ represent unique clip identifiers within the dataset for the specific subject performing the actions (actions 1, 3, and 17, respectively). The intentional randomness of the generated action sequences reflects the unstructured flow of actions encountered in ADL. To further enhance diversity and ensure no bias towards specific subject-action combinations, we shuffled both the action sequences and the subject assignments. This process resulted in the creation of **16,343 stitched videos** with an average 5 actions per video.

**Frame Level Captioning and Dense Descriptions.** This step is the process of generating weak pseudo-labels for automated instruction tuning of the LLVM with the curated dataset. An image captioning model CogVLM [44] is employed to automatically generate frame-level captions for the stitched ADL videos at a rate of $0.5fps$. These captions are subsequently compiled into a dictionary linking each frame identifier to its corresponding description. To enhance the reliability of the pseudo-labels, we implemented an action-conditioned filtering while generating the video descriptions. The dictionary with the frame descriptions, along with the action labels present in the stitched videos, are then used to prompt a GPT 3.5 turbo model to generate a cohesive structured description of the entire stitched video, constrained to a maximum of 300 words. This step leverages the known action labels associated with each video to remove irrelevant noise potentially introduced during the caption generation process. We evaluated various image captioning models, including BLIP-2 [45], and InstructBLIP [46] for frame-level caption generation. However, CogVLM is ultimately chosen due to its ability to generate denser and appropriate descriptions. Please refer to the appendix for our detailed prompting strategy in generating the descriptions.

**Generating QA Pairs.** LLVMs necessitate training data in the form of question-answer (QA) pairs. To generate domain-specific QA pairs for ADL, we leverage the dense video descriptions obtained in the previous step as illustrated in Figure 2. An instruction template (detailed in the Appendix) guides GPT-3.5 in formulating questions across various categories relevant to ADL. These categories include: video summary, performed actions, spatial details, human-object interactions and other video-specific inquiries. Through this prompting approach, we curate a dataset of **100K video instruction pairs**, namely ADL-X, for the stitched ADL videos. These QA pairs benefit from the detailed descriptions and person-centric cropping, resulting in reduced LLM hallucinations compared to other existing methods [17, 20].

Notably, the framework employed for constructing ADL-X from trimmed, labeled action videos can be generalized to other existing datasets. This generalization paves the way for efficient training of domain-specific LLVMs.

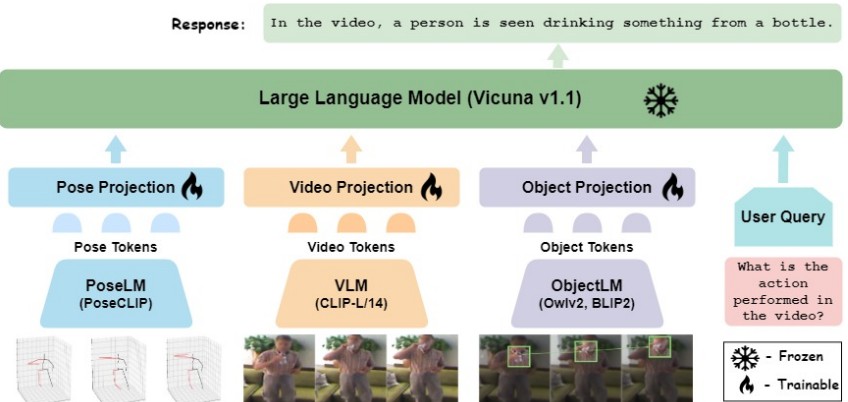

Figure 3: Overview of **LLAVIDAL**, which utilizes an LLM to integrate multiple modalities, including video, pose, and object features. Videos are represented by embeddings obtained from a **VLM**, poses are processed through **(PoseLM)**, and object embeddings are obtained through **(ObjectLM)**. These embeddings are projected into the LLM space, where they are concatenated with tokenized text queries for instruction tuning.

## 3 LLAVIDAL: An LLVM for ADL

LLAVIDAL is a large language vision model designed to align ADL videos with an LLM to generate meaningful conversation about the daily activities performed by humans. This model, similar to Video-ChatGPT [20] and LLaVA [18], integrates a visual encoder with the Vicuna language decoder [47] and is fine-tuned on instructional language-vision data. Unlike Video-ChatGPT [20] and LLaVA [18], LLAVIDAL leverages the random temporal structure present in ADL-X and incorporates additional data modalities such as 3D human poses and human-object interaction cues. This allows LLAVIDAL to generate accurate conversations that are not only contextually appropriate but also temporally aligned with the human activities depicted in the input video. This section will first present a background of LLVM models to align videos with LLMs. Then, we will outline the strategies employed to integrate 3D poses and object interaction cues within the language space of the LLM for enhanced understanding of videos featuring ADL. Subsequently, we will describe the training architecture of LLAVIDAL.

### 3.1 Background: LLVM

Following [20], given an input video denoted by $\nu_i \in \mathbb{R}^{T \times H \times W \times C}$, where $T$ represents the frames encoded using a pretrained vision-language model (**VLM**) CLIP-L/14 [2] to obtain frame-level embeddings for the video, $x_i \in \mathbb{R}^{T \times h \times w \times D}$, with $D$ as the embedding dimension, and $h = H/p$, $w = W/p$ representing the dimensions adjusted by patch size $p$. Temporal and spatial features are extracted by aggregating these frame-level embeddings along the respective dimensions. The video-level features, $V_i \in \mathbb{R}^{F_\nu \times D_\nu}$, are obtained by concatenating the temporal and spatial features, where $F_v$ represents the spatio-temporal tokens and $D_\nu$ is the video feature dimension. The video features are projected into the LLM embedding space using a linear projection layer $\mathcal{T}_v$. Thus, we obtain input tokens $Q_v$ for the video features:

$$Q_v = \mathcal{T}_v(V_i) \in \mathbb{R}^{F_v \times K} \tag{1}$$

The text query is also tokenized such that $Q_t \in \mathbb{R}^{F_t \times K}$. The text query $Q_t$, refers to a question from the training data. The input to the LLM is the concatenation of $Q_t$ and $Q_v$ following the template : [USER: $\langle Q_t \rangle$ $\langle Q_v \rangle$ Assistant:]. We perform instruction-tuning of the LLM on the prediction tokens, using its original auto-regressive training objective. The parameters of the LLM are frozen, thus the loss gradients only propagate through the projection layer $\mathcal{T}_v$.

### 3.2 3D Poses for LLAVIDAL

ADL are rich in actions that primarily involve the movements of critical body parts or joints. The dataset ADL-X includes 3D human poses, which can be utilized to incorporate human kinematics and view-invariant features into the input embedding space of a LLM. These poses can be integrated into the LLM input space in several ways: as an additional text query $Q_t$ for instruction tuning of the LLM, by deriving language descriptions of joint movements to provide context for the LLM, or through features extracted using a suitable pose-language encoder.

**Poses as QA.** We input the 3D joint coordinates alongside the associated human action from the video into GPT-3.5 Turbo [7], which generates a general description of the pose. This description is then re-fed into GPT-3.5 Turbo to generate two QA pairs that provide detailed explanations of the action's motions. These QA pairs are subsequently added to the set of text queries $Q_t$ in our training set for instruction tuning the LLM.

**Poses as Context.** To extract contextual information from human poses, we initially identify five peripheral joints — the head, right hand, left hand, right knee, and left knee — due to their significant contribution to motion in various actions. Using GPT-3.5 Turbo, we generate descriptions of the motion for each of these joints based on their trajectories throughout the video, specifically focusing on how the coordinates of these five joints evolve. The generated descriptions, denoted as $Q_t^p$, are subsequently appended to the text query $Q_t$, incorporates these pose descriptions as additional contextual information. This enriched query $Q_t^{new} = [Q_t^p \ Q_t]$ is then employed for instruction tuning of the LLAVIDAL.

**Poses as Features.** To incorporate poses as tokens into the LLM, it is crucial to align the pose features with a language-contextualized space. To achieve this, we utilize a pretrained Pose-Language model (**PoseLM**), specifically PoseCLIP, to extract pose features that are aligned with the language

domain. The PoseCLIP model comprises a pose backbone [48] and a CLIP text encoder [2], and it undergoes training in two phases. Initially, the pose backbone is pretrained on the NTU RGB+D dataset [49] for action classification. Subsequently, in the second phase, we optimize the similarity between pose features and text features, which encode the prompts describing their action labels, using cross-entropy supervision as outlined in [3]. Further details on the training of this model are provided in the Appendix. These pose features, denoted as $P_i \in \mathbb{R}^{F_p \times D_p}$, where $D_p$ represents the pose feature dimension, can be utilized as input tokens for training LLAVIDAL.

## 3.3 Action-Conditioned Object Cue for LLAVIDAL

To comprehensively understand ADL, it is crucial to not only grasp the semantics of objects but also their trajectories, which are closely linked to the actions performed. Consequently, we propose to explicitly utilize these object trajectories as integral components for training LLAVIDAL. Our framework involves a two-stage pipeline to extract object information directly from RGB video data: (i) *Action-conditioned object detection* and (ii) *Object Localization and Tracking*. Both stages leverage off-the-shelf models that are effective without the need for additional training, facilitating integration into LLAVIDAL for ADL analysis.

**Action conditioned object detection.** Given a stitched ADL video, which comprises a sequence of trimmed video segments (denoted as $clip_j$), the first stage extracts the categories of objects present that are pertinent to the actions performed within each clip. We uniformly sample 8 frames from each video and employ a pre-trained BLIP-2 model [45] to generate a list of distinct objects observed in the frames. To avoid training LLAVIDAL with noisy data, we perform a filtering on the list of objects using the ground-truth action labels and GPT-3.5. Specifically, for each $clip_j$ within a stitched video, we input the corresponding action label and the list of detected objects to GPT-3.5 and prompt it to identify the object(s) most relevant to the given action. For instance, if the objects *plant, chair, bottle, table* are detected in a video labeled with the action *Drinking*, GPT-3.5 is expected to filter out and select [*bottle*] as the relevant object for $clip_j$. Refer to the appendix for our detailed action conditioned object detection prompting strategy.

**Object Localization and Tracking.** Given the list of relevant objects identified in the first stage, the second stage involves spatial localization of these objects within the scene and their temporal association (i.e., object tracking) based on the feature similarity of the image regions corresponding to the localized objects in the stitched video. We employ a pre-trained open vocabulary object localization model (**ObjectLM**), OWLv2 [50], and input the list of relevant objects detected in stage 1 along with the corresponding video. Localization and tracking are performed on 8 frames that are uniformly sampled from $clip_j$ within a stitched video. For each frame, we obtain bounding boxes $B_t \in \mathbb{R}^{n \times 4}$, where each bounding box corresponds to one of the $n$ relevant objects in the $t$th frame. Features for each object are then extracted from the image regions within these bounding boxes using our object localization model. We denote the features for the objects in frame $t$ as $O_t \in \mathbb{R}^{8n \times D_o}$, where $D_o$ is the object feature dimension. To associate objects across frames, we utilize a feature-based object tracking approach. Specifically, for each object in frame $t$, represented by the feature vector $O_i^t \in \mathbb{R}^{D_o}$, we compute the cosine similarity between $O_i^t$ and all feature vectors in frame $t + 1$. The object $i$ in frame $t$ is then associated with the object in frame $t + 1$ that exhibits the highest similarity score. This matching process is iterated for all objects in each frame, thereby establishing a track for each relevant object throughout the sampled frames. These object tracks, with corresponding bounding boxes and features, facilitate the integration of object information into the training of LLAVIDAL: Object as QA, Object as context, and Object as features.

**Object as QA.** Similar to the approach taken with poses, to generate QA pairs for objects, we formulate a question based on the trajectory coordinates of the relevant object(s). These QA pairs are added to the set of text queries $Q_t$ for instruction tuning LLAVIDAL.

**Object as Context.** To integrate the context of detected objects into the LLM space, we append the list of relevant object labels, denoted by $Q_t^o$, to each text query token $Q_t$. Consequently, the updated text query is represented as $Q_t^{new} = [Q_t^o \ Q_t]$. This enhanced text query, $Q_t^{new}$, is utilized for instruction tuning.

**Object as Features.** The object features extracted during the object localization and tracking stage are utilized as input tokens $Q_o \in \mathbb{R}^{8n \times D_o}$, which are incorporated alongside the text query tokens ($Q_t$) and input video tokens ($Q_v$). For $n$ relevant objects detected, the object query $Q_o$ is structured using

the following template $[\langle Q_o \rangle = \langle Q_o^1 \rangle \langle Q_o^2 \rangle ... \langle Q_o^n \rangle]$ where $Q_o^j \in \mathbb{R}^{8 \times D_o}$ represent the features of each relevant object in the video.

## 3.4 Training LLAVIDAL

As illustrated in Figure 3, the QA pairs, along with context or features obtained from the RGB video, 3D poses, and object cues can be integrated into LLAVIDAL. Integrating QA pairs and contextual information is straightforward; they are introduced into $Q_t$ and trained using standard methods for LLVM. However, to integrate other modalities with features, we feed these additional cues through specific projection layers designed to align them with the input space of the LLM. Accordingly, the video, pose, and object features are projected into the LLM embedding space using linear projection layers $\mathcal{T}_j$ for each cue $j = \{v, p, o\}$, resulting in LLM input token representation of the video, pose, and object cues, respectively:

$$Q_v = \mathcal{T}_v(V_i); \quad Q_p = \mathcal{T}_p(P_i); \quad Q_o = \mathcal{T}_o(O_i) \tag{2}$$

where $Q_j \in \mathbb{R}^{F_j \times K}$. Thus, the input to the LLM comprises the concatenation of $Q_t$ and $Q_j$ for $j = \{v, p, o\}$, structured according to the template: [USER: $\langle Q_t \rangle \langle Q_v \rangle \langle Q_o \rangle \langle Q_p \rangle$ Assistant:]. This training scheme ensures that the video, object, and pose cues are effectively aligned to the LLM embedding space, facilitating an accurate understanding of ADL. During the **inference**, LLAVIDAL utilizes only the holistic video cue, omitting person-centric cropping and consequently eliminating additional cues. In practice, the embedding dimensions are $D_v = 1024$ for visual, $D_o = 512$ for object features, $D_p = 216$ for pose features and $K = 4096$. The number of tokens is set as $F_v = 356$ and $F_p = 256$ for visual and pose tokens respectively. We train LLAVIDAL for 3 epochs with a batch size of 32 and a learning rate of $2e^{-5}$ on 8 A6000 48GB GPUs. For the purpose of promoting research in this field, we also provide the pose features and object trajectories of LLAVIDAL along with the dataset.

## 4 Experiments

### 4.1 Experimental Setting

**Evaluation Metrics.** Inspired by [20], LLVM's ability to generate video-level descriptions is evaluated. This involves comparing the generated descriptions with ground truth and scoring them on dimensions such as Correctness of Information, Detail Orientation, Contextual Understanding, Temporal Understanding, and Consistency, with scores scaled to be bounded at 100. Due to the subjective nature of this metric, Mementos Evaluation [51] is also conducted to assess the recognition of common action-verbs and object-nouns in the video descriptions compared to ground truth, presenting F1 scores for these classifications. However, comparing video descriptions generated by LLVMs presents a challenge due to the inherently subjective nature of these descriptions. Some objective evaluation benchmarks for LLVMs [52, 53, 54] primarily focus on video tasks involving in-the-wild activities. Therefore, this paper introduces novel benchmarks for assessing LLVM's temporal understanding of ADL videos. We propose two new **ADLMCQ** benchmarks including ADLMCQ-AR and ADLMCQ-AF. ADLMCQ-AR involves multiple-choice question-answering for action recognition, where the model selects the correct action from a set of options given a question about the action performed in a video. Similarly, ADLMCQ-AF focuses on action forecasting, requiring the model to predict the next action based on the preceding actions. It is important to note that all evaluations are performed zero-shot.

**Evaluation Datasets.** For ADLMCQ-AR evaluation, we utilize the Charades [55] and Toyota Smarthome [56] datasets. Evaluation for ADLMCQ-AF is conducted using LEMMA [57] and Toyota Smarthome Untrimmed (TSU) [58] datasets. Video description tasks are assessed using the Charades and TSU datasets, both featuring long-duration videos with multiple actions per video. Notably, for the TSU dataset, we manually annotated video descriptions with fine-grained details regarding activities performed by elderly individuals, employing 6 human annotators for 174 videos. Our evaluation relies on these annotated descriptions, which we also provide to the community as part of the test set for ADL-X.

### 4.2 Impact of ADL-X Training on LLVMs

To understand the requirement of ADL-X, we assess VideoChatGPT [20] trained on 100K instruction pairs from ActivityNet [25], trimmed NTU120 [39], and ADL-X in Table 2. Notably,

Table 2: Impact of ADL-X Training

| Method | Training Data | ADLMCQ-AR (Smarthome) | ADLMCQ-AF (LEMMA) | Action Description (Charades) | | |
|---|---|---|---|---|---|---|
| | | | | Object | Action | Correctness |
| VideoChatGPT [20] | ActivityNet | 40.8 | 35.7 | 14.8 | **16.1** | 35.8 |
| VideoChatGPT [20] | NTU120 | 49.8 | 33.5 | 27.0 | 10.1 | 38.8 |
| ADL-X ChatGPT [20] | ADL-X | **52.3** | **44.8** | **32.2** | 13.4 | **43.0** |

ADL-X ChatGPT, trained on ADL-X, consistently outperforms the others in both ADLMCQ-AR and ADLMCQ-AF tasks. However, it's worth mentioning that while the baseline [20] exhibits strong performance in the action metric of Mementos, it notably underperforms in the object metric. It's important to emphasize that ADLMCQ evaluations offer more objective and reliable assessments for understanding the temporal comprehension of LLVMs.

Table 3: Introducing Pose and Object Cues into LLAVIDAL

| Method | ADLMCQ-AR | | ADLMCQ-AF | | AD (Charades) | | AD (TSU) | |
|---|---|---|---|---|---|---|---|---|
| | Charades | Smarthome | LEMMA | TSU | Object | Action | Object | Action |
| ADL-X ChatGPT | 58.0 | 52.3 | 44.8 | 25.25 | 16.6 | 14.8 | 16.6 | 14.8 |
| Pose QA | 48.5 | 49.0 | 42.0 | 21.2 | 31.8 | 14.0 | 16.5 | 15.9 |
| Pose Context (PC) | 50.8 | 54.0 | 45.0 | 22.3 | 30.5 | **14.8** | 18.6 | 15.4 |
| Pose Features (PF) | 56.7 | **57.0** | **51.3** | 26.0 | **32.7** | 13.5 | 18.2 | 13.0 |
| PC + PF | 52.5 | 53.1 | 44.6 | 24.9 | 32.1 | 13.6 | 17.5 | 15.6 |
| Object QA | 51.1 | 50.1 | 40.3 | 23.0 | 32.1 | 13.7 | 17.0 | 16.0 |
| Object Context | 44.6 | 46.2 | 41.8 | 21.0 | 31.2 | **14.7** | 17.2 | 16.5 |
| Object Features (OF) | **59.0** | **58.8** | 52.6 | 27.0 | 33.1 | 14.3 | 18.0 | **17.7** |
| PF + OF | 56.2 | 56.1 | 51.0 | 26.6 | 30.4 | 14.1 | **20.0** | 14.1 |

### 4.3 How to introduce object and pose cues into the LLM space?

Table 3 explores the integration of pose and object cues into LLAVIDAL. We evaluate incorporating poses as QA, context (PC), and features (PF). While both pose context and features outperform the baseline ADL-X ChatGPT, projecting pose features directly into the LLM embedding space yields superior performance. This suggests the effectiveness of language contextualization for pose information. Combining pose context and features hinders performance, suggesting potential redundancy. In contrast, object cues as QA or context offer minimal discriminative information for the LLM. However, object features derived from ObjectLM significantly improve performance across most tasks, highlighting their importance in understanding ADL. A detailed analysis of these cues' impact on ADLMCQ action classes is provided in the Appendix, revealing complementary information learned. Interestingly, LLAVIDAL with object features outperforms the model with pose features on all tasks. However, attempts to combine both pose and object features result in performance converging towards the pose-only model. We hypothesize this is due to the challenge of optimizing the projection layer $\mathcal{T}_v$ that effectively aligns both $\mathcal{T}_p$ and $\mathcal{T}_o$. Therefore, multi-cue integration is left for future work. Given its superior performance, LLAVIDAL with object features is used for the remainder of the paper.

Table 4: Performance on Video Description. [CI: *Correctness of Information*, DO: *Detail Orientation*, CU: *Contextual Understanding*, TU: *Temporal Understanding*, Con: *Consistency*]

| Method | Training Data Size | Charades | | | | | | | TSU | | | | | | |
|---|---|---|---|---|---|---|---|---|---|---|---|---|---|---|---|
| | | Object | Action | CI | DO | CU | TU | Con | Object | Action | CI | DO | CU | TU | Con |
| CogVLM [44] + GPT [7] | 1.5B Images | 19.8 | 9.4 | 44.2 | 42.0 | 33.2 | 33.0 | 40.6 | 16.8 | 6.1 | 41.0 | 37.0 | 37.6 | 34.4 | 40.2 |
| CogVLM [44] + Llama [11] | 1.5B Images | 20.9 | 9.3 | 44.2 | 41.8 | 34.8 | 32.0 | 40.6 | 17.9 | 7.8 | 30.0 | 33.4 | 35.4 | 33.8 | 30.0 |
| BLIP2 [45] + GPT [7] | 1.5B Images | 21.1 | **17.3** | 33.6 | 33.8 | 35.4 | 30.0 | 34.4 | **23.2** | 22.8 | 38.0 | 35.4 | 30.6 | 37.2 | 38.4 |
| VideoLlama [19] | 2.6M QA Pairs | 14.7 | 15.9 | 32.2 | 32.0 | 36.0 | 34.4 | 39.6 | 21.0 | 13.4 | 33.2 | 30.4 | 31.2 | 34.6 | 42.0 |
| VideoLlava [18] | 1.2M QA Pairs | 15.8 | 15.5 | 38.2 | 44.4 | **44.0** | 37.4 | 40.2 | 20.9 | 15.3 | 37.8 | 33.8 | 40.2 | 40.4 | 39.6 |
| VideoChatGPT [20] | 100K QA Pairs | 14.8 | 16.1 | 35.8 | 44.2 | 41.6 | 42.2 | 37.8 | 21.8 | 18.0 | 43.0 | 45.8 | 41.4 | 43.0 | 50.0 |
| ADL-X ChatGPT [20] | 100K QA Pairs | 32.2 | 13.4 | 43.0 | 46.8 | 42.2 | 43.8 | 38.6 | 16.6 | 14.8 | 43.0 | 47.2 | 39.6 | 37.6 | 50.0 |
| LLAVIDAL | 100K QA Pairs | **33.1** | 14.3 | **51.8** | **54.2** | **44.0** | **49.2** | **41.8** | 18.0 | 17.7 | **46.0** | **48.6** | **42.2** | **45.8** | **58.0** |

### 4.4 Comparison to the state-of-the-art

We compare LLAVIDAL against the state-of-the-art (SOTA) in the performance on video description generation and ADLMCQ tasks involving action recognition and forecasting.

**Video Description Generation.** Table 4 shows the performance comparison of baseline LLVMs and LLAVIDAL on their video description capabilities on the Charades and TSU datasets. Video-level descriptions are obtained directly from the Charades dataset. For the TSU dataset, comprising lengthy videos, we segment each video into 1-minute clips and input them individually to the LLVMs for

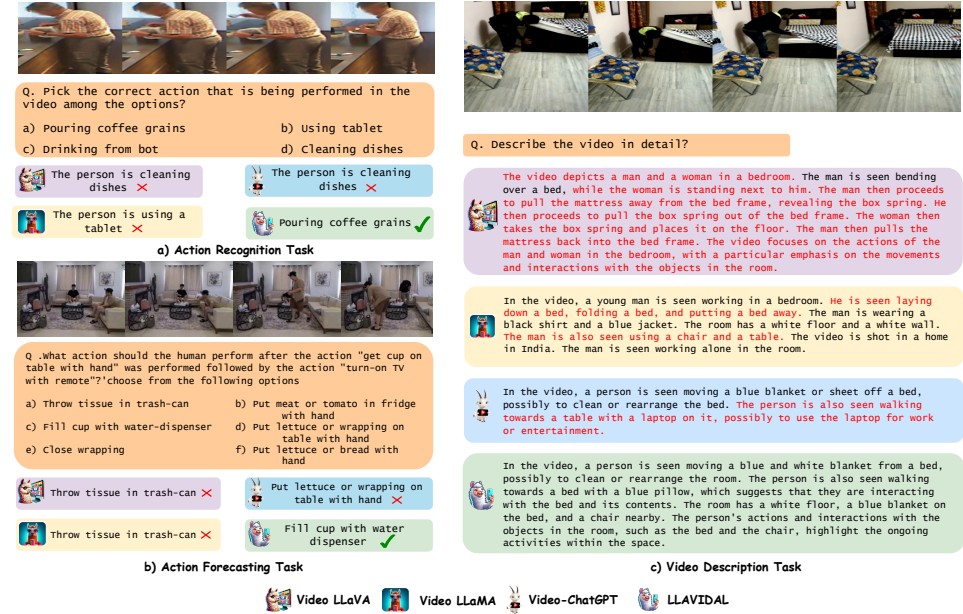

Figure 4: Qualitative results comparing LLAVIDAL with SOTA models. Incorrect descriptions are marked in red.

generating clip-level descriptions. Subsequently, we concatenate all clip-level descriptions and utilize GPT-3.5 turbo to summarize them into a video-level description, following the same instruction template utilized in our dense description pipeline for ADL-X. LLAVIDAL consistently surpasses SOTA and outperforms all models including, image captioners-summarizers pipelines which are trained on billions of images, across all 5 VideoChatGPT metrics. However, in the Mementos Evaluation, LLVM baselines exhibit superior performance over LLAVIDAL in the Smarthome domain. This discrepancy may be attributed to the loss of relevant information when generating video-level descriptions using GPT.

Table 5: ADLMCQ - Action Recognition

| Method | Charades | Smarthome |
|---|---|---|
| VideoLlama [19] | 33.0 | 27.4 |
| VideoLlava [18] | 44.4 | 54.0 |
| VideoChatGPT [20] | 56.0 | 40.8 |
| ADL-X ChatGPT [20] | 58.0 | 52.3 |
| LLAVIDAL | **59.0** | **58.8** |

Table 6: ADLMCQ - Action Forecasting

| Method | LEMMA | TSU |
|---|---|---|
| VideoLlama [19] | 20.8 | 15.6 |
| VideoLlava [18] | 32.2 | 20.2 |
| VideoChatGPT [20] | 35.7 | 25.0 |
| ADL-X ChatGPT [20] | 44.8 | 25.3 |
| LLAVIDAL | **52.6** | **27.0** |

**ADLMCQ.** Table 5 compares LLAVIDAL to SOTA LLVMs on the ADLMCQ-AR benchmark. LLAVIDAL achieves significant improvements, surpassing VideoChatGPT by +5.4% and +44.1% on the Charades and Smarthome datasets, respectively. Similarly, Table 6 demonstrates LLAVIDAL's superiority on the ADLMCQ-AF benchmark. It outperforms VideoChatGPT by up to +47.3%, highlighting its exceptional capability in action forecasting tasks.

Figure 4 provides a visual comparison of LLAVIDAL against representative baselines on the ADL benchmarks. More visual samples are provided in the Appendix.

## 5 Conclusion & Future Work

In this work, we present a framework for curating ADL datasets for instruction tuning LLVMs, thus introducing ADL-X. We introduce LLAVIDAL, an LLVM capable of integrating 3d poses and human-object interaction cues by projecting their language contextualized representations into the LLM embedding space. To assess LLVM performance in ADL scenarios, we propose the ADLMCQ benchmark. Results demonstrate that LLAVIDAL, when trained on ADL-X, surpasses other LLVM baselines in ADLMCQ tasks, indicating its efficacy in grasping intricate temporal relationships within ADL contexts. Future research will focus on expanding ADL-X by integrating additional curated ADL datasets and exploring stage-wise training strategies to effectively integrate both pose and object cues within LLAVIDAL.

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
