# OpenReview forum: "LLAVIDAL: Benchmarking \underline{L}arge \underline{LA}nguage \underline{VI}sion Models for \underline{D}aily \underline{A}ctivities of \underline{L}iving"
_NeurIPS.cc/2024/Datasets_and_Benchmarks_Track — Submitted to NeurIPS 2024 Track Datasets and Benchmarks_

### Official Review · Reviewer_Edmn · 2024-07-21
**A novel ADL-X training set is proposed and a novel LLAVDIAL model is fine-tuned on ADL-X to improve performance of LLVM on DAL.**

**Rating:** 7
**Confidence:** 3
**Correctness:** The claims made in the submission are…
**Clarity:** The paper is well written.

**Review:**

Pros:
1. This paper observed the performance gap of current LLVM on daily activities of living videos.
2. This paper creates a novel ADL-X dataset based on many existing tools and existing datasets to improve the performance of LLVM on daily activities of living videos. The methods proposed to create this dataset is scalable.
3. Thorough experiments are carried out to find the best way to make use of new signals introduced, such as human pose, action related object and their location and trajectory, which makes the design of instruction tuning of LLAVDIAL convincing.
4. The proposed LLAVDIAL outperforms exiting video base LLVM models.
5. The paper is well written and easy to follow.
Cons:
1. LLAVIDAL is trained extened with OF in the final design. One design that is not clear to me is whether OF is used during the inference of evaluation. LLAVIDAL be performed without OF to save compute from object detection and tracking?

**Strengths:**

The paper proposed both a training dataset and instruction tuning recipe to improve the performance of LLVM on daily activities of living videos. It makes use of new signals such as human pose, action related object and their location and trajectory to improve the LLVM which is sound to me. The proposed LLAVDIAL model out performs existing models on Charades and Smarthome dataset.

**Additional Feedback:**

1. Line 35, "as shown in 1" should this be Figure 1?

**Documentation:**

Documentation has sufficient details.

**Ethics:**

I don't have ethics concerns.

**Limitations:**

The authors adequately addressed the limitations and potential negative societal impact of their work.

**Opportunities For Improvement:**

Please refer to review section.

**Relation To Prior Work:**

Related works are clearly discussed.

**Summary And Contributions:**

This paper proposed a novel ADL-X dataset which includes human pose, action related object and their location and trajectory besides descriptions of the video to improve the performance of ability of LLVM on understanding daily activities of living videos. Based on this dataset, they also instruction fine-tuned a LLVM called LLAVDIAL which achieves the state-of-the-art performance on  daily activities of living videos.

---

> ### Author Rebuttal · Authors · 2024-08-17
>
> We thank Reviewer EDMM for their encouraging comments and insightful feedback on our work.
>
> ---
>
> >Does LLAVIDAL use OF during inference? Can inference be performed without OF to save compute from object detection and tracking?
>
> We clarify that LLAVIDAL does not use OF at inference time (see lines 261-263). The reviewer is correct that performing inference using OF would incur additional computational costs, motivating our use of OF exclusively during training. We refer the reviewer to Figure 4 in the rebuttal PDF for a runtime analysis of LLAVIDAL. In this figure, we show the average FPS of LLAVIDAL inference with and without OF relative to performance. This figure shows that including OF at inference leads to a slight increase in performance with a major tradeoff of ~5x decrease in FPS.
>
> ---
>
> **Typo on line 35**. We appreciate the reviewer’s attention to detail, we will thoroughly review the manuscript and correct this typo in the final version.

---

### Official Review · Reviewer_56PD · 2024-08-03
**Daily activity datasets with diverse modalities, but paper writing needs improvement**

**Rating:** 4
**Confidence:** 3
**Correctness:** Correct
**Clarity:** Some places lacks clarity, see more d…

**Review:**

Pro:
1) The proposed datasets contain a diversity of modalities
2) The paper performed ablation studies on how to better integrate some modalities into the vision language model.

Con:
1) The paper writing quality needs to be improved.

    a)  It's unclear what defines video instruction dataset, or how the proposed dataset differ from Ego4D, HowTo100M, EpicKitech, which are also about daily activities

    b)  The paper doesn't discuss related work in the main paper, for example, how datasets or models in this field.

    c)  The paper doesn't introduce clearly the structure and format of the dataset in main paper not appendix

2) The novelty is incremental given the approach is similar to prior works [1][2][3], but constrained to a more specific domain.

[1] Sequential Modeling Enables Scalable Learning for Large Vision Models https://arxiv.org/pdf/2312.00785
[2] Unified-IO 2: Scaling Autoregressive Multimodal Models with Vision, Language, Audio, and Action https://arxiv.org/pdf/2312.17172
[3] PaliGemma: A versatile 3B VLM for transfer https://arxiv.org/pdf/2407.07726

**Strengths:**

1) The proposed datasets contain a diversity of modalities
2) The paper performed ablation studies on how to better integrate some modalities into the vision language model.

**Additional Feedback:**

None

**Documentation:**

Needs better documentation on the data organization.

**Ethics:**

No concern

**Limitations:**

No such limitations

**Opportunities For Improvement:**

1) The paper writing quality needs to be improved.

    a)  It's unclear what defines video instruction dataset, or how the proposed dataset differ from Ego4D, HowTo100M, EpicKitech, which are also about daily activities

    b)  The paper doesn't discuss related work in the main paper, for example, how datasets or models in this field.

    c)  The paper doesn't introduce clearly the structure and format of the dataset in main paper not appendix

2) The novelty is incremental given the approach is similar to prior works [1][2][3], but constrained to a more specific domain.

[1] Sequential Modeling Enables Scalable Learning for Large Vision Models https://arxiv.org/pdf/2312.00785
[2] Unified-IO 2: Scaling Autoregressive Multimodal Models with Vision, Language, Audio, and Action https://arxiv.org/pdf/2312.17172
[3] PaliGemma: A versatile 3B VLM for transfer https://arxiv.org/pdf/2407.07726

**Relation To Prior Work:**

Not well discussed in the main paper, but discussed in the supplementary

**Summary And Contributions:**

The paper's contributions
1) curates a multi-view instruction dataset for daily activities
2) develops a model based on finetuning language vision models.
3) proposes a benchmark to evaluate the capability of understanding daily activities

---

> ### Author Rebuttal · Authors · 2024-08-17
>
> We thank Reviewer 56PD for their comments and feedback on our work.
>
> ---
>
> >Video-instruction dataset was never defined
>
> We clarify that a video-instruction dataset comprises paired instructional videos and corresponding semantics (such as language or human skeletons), where the semantics provide context, explanation, or instruction relevant to the visual content. These datasets have become increasingly common in the video understanding domain. For further details, we refer the reviewer to the Related Work section of Miech et al. (Howto100M, ICCV 2019). In response to the reviewer’s concern, we will formally define this term in the final manuscript.
>
> ---
>
> >It’s unclear how the proposed dataset differ from Ego4D, HowTo100M, EpicKitchens, which are also about daily activities
>
> We would like to clarify that this work focuses exclusively on Activities of Daily Living (ADL) recorded from a third-person viewpoint, as detailed in lines 23-26 of the manuscript. Since Ego4D and EpicKitchens are recorded from a first-person (egocentric) perspective, they do not represent the evaluation setting of our work. The egocentric nature of datasets like Ego4D and EpicKitchens distinguishes them from our proposed exocentric ADL-X dataset.
>
> We respectfully disagree with the reviewer that HowTo100M is representative of the types of activities of daily living targeted in this work. HowTo100M is a large dataset of “narrated instructional videos” collected from YouTube, which differs from our proposed setting. Our work specifically addresses the limitations of Large Language Vision Models (LLVMs) on real-world activities of daily living performed in indoor scenarios, as discussed on lines 44-46. The distinction between YouTube data and real-world data is also emphasized in Miech et al. (HowTo100M, ICCV 2019), where Table 1 differentiates Charades and EpicKitchens as being recorded in a “Home” environment, in contrast to the “YouTube” environment of HowTo100M.
>
> ---
>
> >The paper discusses related works in the supplementary and not main paper
>
> We plan to relocate this section to the main paper in the final version given the additional page. We refer the reviewer to our general response for a detailed discussion regarding how we plan to address this concern.
>
> ---
>
> >The paper introduces the structure and format of the dataset in the supplementary rather than the main paper
>
> In the interest of space, many of these details were moved from the main paper to the supplementary. Following the reviewer’s feedback, we will shift the dataset statistics from Section D in the supplementary to the main paper. If there are other specific dataset details the reviewer is referring to we are happy to clarify them in the main paper.
>
> ---
>
> >The novelty is incremental given the approach is similar to prior works [1][2][3], but constrained to a more specific domain
>
> We clarify that this work presents a framework for curating ADL datasets specifically for instruction tuning LLVMs, introducing the ADL-X dataset, creating new benchmarks for ADL tasks, and developing the LLAVIDAL model, which surpasses baseline performance on these tasks. We believe these contributions align well with the Dataset & Benchmark track of NeurIPS.
> Additionally, we respectfully disagree with the reviewer’s assessment that LLAVIDAL lacks novelty compared to the prior works mentioned, and we provide further clarifications below.
>
> **Constrained to a specific domain**. We agree with the reviewer that LLAVIDAL is designed with a specific focus on understanding ADL. However, we view this as a strength rather than a limitation, as our goal in this work is to develop a domain-specific LLVM tailored to ADL scenarios. By concentrating on ADL, LLAVIDAL is able to acquire a nuanced and detailed understanding of this complex domain, which is not achievable with more general-purpose approaches.
>
> **Incremental compared to prior works**. While works such as [1,2,3] are similar to LLAVIDAL in their integration of multiple modalities into a unified framework, the key distinction lies in the downstream tasks they are designed for. None of these approaches are explicitly designed to tackle the challenges inherent in ADL. The novelty of LLAVIDAL lies in its use of unique modalities, such as 3D human poses and object tracks, to train LLVMs—modalities that are not explored in generic LLVMs [1,2,3], yet are crucial for capturing the intrinsic complexities of ADL.
>
> ---
>
> >Needs better documentation on the data organization
>
> We appreciate the reviewer's concern and consequently updated the README file in the git repository. We have made the following changes to the structure of our repository:
> - We have added a table to the beginning of the README containing links to all of the relevant data download, model downloads, and training details
> - We have added a section to the README detailing 1) the technical format of the ADL-X instruction tuning dataset and 2) the structure and technical format of the multimodal features of ADL-X
>
> We recognize the importance of well documented code repositories to enable reproducibility, and we hope that these changes are a step towards that.

---

> > ### Author Response · Authors · 2024-08-25
> >
> > With one week remaining in the discussion period, we are wondering if we can provide any additional clarifications to the reviewer's concerns or if the current rebuttal has addressed them sufficiently.
> >
> > Once again we thank the reviewer for their time spent reviewing our manuscript.

---

### Official Review · Reviewer_SD4k · 2024-08-07
**Potentially useful data augmentation technique**

**Rating:** 6
**Confidence:** 3
**Clarity:** The paper is clear and well-written.

**Review:**

The main motivation for datasets that specifically cater to daily activities is due to the large domain shift between large scale web video datasets that are typically interesting events like sports, animal videos, movie excerpts, instructional videos, etc.

The paper further emphasizes the need for understanding of the 3D spatial position of the persons performing various complex daily activities. To this end, it proposes a semi-automated framework for curating and training large VLMs.

The paper validates the usefulness of the dataset with experiments with their proposed model that integrates multiple data sources. The proposed model LLAVIDAL, is basically an integration of multiple modalities of data. Despite not being a novel architecture, it demonstrates improvement when training with ADLX.

There are many details and design decisions in the semi-automatic labeling process that would be good to mention / discuss in the paper draft.

**Strengths:**

- Really large dataset containing 100K video instruction pairs. The semi-automatic method to construct that dataset is also quite general, and could potentially be replicated for other datasets that could benefit from additional instruction labels. The synthetic data generation approach leverages existing open-source VLMs and LLMs, and results in large-scale data that can be generated at a fraction of the cost of human data.

- The approach integrates information from multiple modalities -- pose features, object features, image (video) features. Lightweight projection layers appear to be sufficient to align features from these disparate modalities, with a frozen pre-trained Vicuna LLM decoder.

- Experiments demonstrate that the ADLX dataset significantly improves performance of LLAVIDAL on multiple tasks.

**Additional Feedback:**

None

**Correctness:**

The motivation for the dataset + approach is clear.

The semi-automatic synthetic data generation approach involves multiple models that likely each introduce some error. This is a limitation of the dataset. However, this could be utilized for augmenting a training dataset.

**Documentation:**

Code is available. The dataset seems to be available behind a micrsoft sign-in (which didn't work for me).

**Limitations:**

See "opportunities for improvement"

**Opportunities For Improvement:**

The approach primarily utilises pretrained models to perform feature extraction of pose, image, and object features. Descriptions from GPT3.5 are generated directly based on the human action and joint co-ordinates. It's unclear to what extent the pretrained GPT models are capable of understanding the semantic information conveyed by the pose skeleton that is in motion, given simply the skeleton's joint co-ordinates. It would be nice to represent the  skeleton in an alternate format whose semantics could be more readily processed by a language model.

It would also be nice to see example intermediate results / generations from each of the components (e.g., above).

**Relation To Prior Work:**

Very briefly discussed.

**Summary And Contributions:**

- The paper proposes a framework for curating ADL multi-view datasets, and creates ADL-X with the goal of fine-tuning large VLMs. ADL-X contains 100K RGB with (video, instruction) pairs, language descriptions, 3D skeletons (point), action-conditioned object trajectories. When trained on ADLX, the proposed model LLAVIDAL achieves SotA across all ADL eval metrics

- The paper also implements a novel multi-modal model, LLAVIDAL, which is capable of incorporating 3D poses, and relevant object trajectories. The motivation behind LLAVIDAL is to synthesize data from different modalities, and perform higher-level reasoning regarding spatiotemporal relationships between objects, persons, etc. in daily activities

- Finally, the paper proposes the ADLMCQ multiple-choice task regarding daily activities which includes action recognition and action forecasting. It provides an easy setting for evaluation of the LLMs’ ability to understand daily activities

---

> ### Author Rebuttal · Authors · 2024-08-17
>
> We thank Reviewer SD4k for their thoughtful comments and insightful feedback on our work.
>
> ---
>
> >It's unclear to what extent the pretrained GPT models are capable of understanding the semantic information conveyed by the pose skeleton
>
> We agree with the reviewer that pretrained GPT models may struggle to capture the motion semantics in relation to actions when prompted simply with skeleton coordinates. This is evident from our “Pose as QA” approach, where GPT3.5 is prompted with skeleton joint coordinates to generate video-instruction pairs (more details on lines 177-181). Table 3 shows that this approach performs suboptimally to the baseline in many cases, indicating the inability of pretrained GPT models to capture the relevant motion semantics. However, we find that GPT can provide meaningful motion semantics when they are used to augment the text query. This is realized in our “Pose as Context” approach described on lines 182-189. Table 3 demonstrates its consistent performance over “Pose as QA” and the baseline.
>
> **Alternative skeleton formats**. Regarding alternative formats for representing the skeleton data, we suggest that this alternative format can be realized as our “Pose as Features” approach discussed on lines 190-199. This approach provides LLAVIDAL with motion semantics obtained from a pose-language model rather than natural language obtained from LLMs prompted with joint coordinates, and its superiority over language-based approaches is demonstrated in Table 3.
>
> **Summary**. These results suggest that while prompting strategies can help GPT’s ability to capture meaningful semantics from skeleton data, alternative representation learning based approaches provide a richer form of skeleton supervision than natural language-based approaches.
>
> ---
>
> >It would also be nice to see example intermediate results / generations from each of the components
>
> Please see Figures 1,2,3 in the rebuttal PDF for the intermediate results of Pose as QA and Pose as Context; Object as QA and Object as context; and video only QA. In these figures, we have highlighted the disparities in text generated by LLAVIDAL for various queries to illustrate its performance across different prompting strategies. For example in Figure 1, we see Pose as Context better captures the salient motion details relevant to the action.
>
> We plan to include these figures with the intermediate results in the supplementary material of the final manuscript.
>
> ---
>
> **More details and design decisions of automatic labeling**. In the interest of space, many of these details were moved from the main paper to the supplementary material (see section H in Appendix). We would appreciate knowing if there are any other specific details or decisions the reviewer would like to see, and we are happy to clarify them in the manuscript.
>
> ---
>
> **Prior work is only very briefly discussed**. We note that an extensive related work section is made available in the supplementary and refer the reviewer to our general response for a detailed discussion regarding how we plan to address this in the final version of the manuscript.
>
> ---
>
> >The dataset seems to be available behind a microsoft sign-in (which didn't work for me)
>
> We apologize for this issue in our dataset hosting. We have fixed this issue by shifting our dataset to HuggingFace, and we have updated our website and Github repository accordingly.

---

### Author Rebuttal · Authors · 2024-08-17

We thank all reviewers for their thoughtful and constructive comments on our work.

We are glad to see that two of the three reviewers recommended acceptance, and encouraged that reviewers find strength in the contribution of our dataset (SD4k, 56PD, EDMM), thorough experimentation of our method (SD4k, EDMM), ablation studies (56PD), paper writing (SD4k, EDMM), as well as the convincing design (EDMM), and superior performance of LLAVIDAL on activities of daily living (SD4k, EDMM).

We provide a general response to the concern shared by multiple reviewers here, and provide individual responses to each reviewer separately.

---

**1. The main paper does not discuss related work in enough detail**

We thank Reviewers SD4k and 56PD for raising this concern. In the interest of space we decided to shift the related works section from the main paper to the supplementary materials. Given the additional content page in the camera ready version, we will provide a distilled version of the related works in the main paper to address the concerns of the two reviewers.

---

### Author Response · Authors · 2024-08-25

As there is only one week remaining in the discussion period, we are happy to discuss any reviewer's concerns that remain unanswered or unclear after our initial response.

We thank all reviewers for their time spent reviewing our rebuttal and manuscript.

---

### Author Response · Authors · 2024-08-30
**Please Acknowledge the Rebuttal Before the Discussion Period Ends**

With only one day remaining in the discussion period, we request that the reviewers at least acknowledge the rebuttal. This will provide us a fair opportunity to address any additional concerns you may have.

Thank you.

---

### Decision · Program_Chairs · 2024-09-26

**Decision:**

Reject

**Comment:**

The paper introduces a framework for curating ADL multi-view datasets and presents ADL-X, designed to fine-tune large Vision-Language Models (VLMs). ADL-X comprises 100,000 RGB images paired with video, instruction, language descriptions, 3D skeletons, and action-conditioned object trajectories. When trained on ADL-X, the proposed model, LLAVIDAL, achieves state-of-the-art performance across all ADL evaluation metrics.

Reviewers agree on the dataset's significance for advancing research in human daily activity understanding. However, one reviewer raises concerns about the similarity to existing datasets and methods. The authors have not fully addressed this concern. Considering its limited novelty, the authors need to further improve the submission.